A new glimpse on Mesozoic zooplankton—150 million-year-old lobster larvae

Haug Joachim T. joachim.haug@palaeo-evo-devo.info
Haug Carolin
Department of Biology, Ludwig-Maximilians-Universität München , Planegg-Martinsried , Germany
GeoBio-Center, Ludwig-Maximilians-Universität München , Germany
De Baets Kenneth
Electronic publication date: 2017 Feb 2
Publication date: 2017
Volume: 5
Electronic Location ID: e2966
Received 2016 Sep 27; Accepted 2017 Jan 7
Copyright: ©2017 Haug and Haug
Copyright year: 2017
Copyright holder: Haug and Haug
License: This is an open access article distributed under the terms of the Creative Commons Attribution License, which permits unrestricted use, distribution, reproduction and adaptation in any medium and for any purpose provided that it is properly attributed. For attribution, the original author(s), title, publication source (PeerJ) and either DOI or URL of the article must be cited.
License URL: https://creativecommons.org/licenses/by/4.0/

Keywords: Nephropida, Upper jurassic, Homarida, Zoea, Solnhofen

Funding: German Research Foundation DFG HA 6300/3-1 DFG HA 7066/3-1) LMUexcellent investment fund Integrated Infrastructure Initiative programme SYNTHESYS FR-TAF-5175 FR-TAF-5181 Joachim Haug and Carolin Haug were supported by the German Research Foundation (DFG HA 6300/3-1, DFG HA 7066/3-1). JTH received support via the LMUexcellent investment fund. The research visits at MNHN Paris of CH and JTH have been made possible by grants from the European Commission’s (FP 6) Integrated Infrastructure Initiative programme SYNTHESYS (FR-TAF-5175, FR-TAF-5181). The funders had no role in study design, data collection and analysis, decision to publish, or preparation of the manuscript.

==============================
Larvae of malacostracan crustaceans represent a large fraction of modern day zooplankton. Plankton is not only a major part of the modern marine ecosystem, but must have played an important role in the ecosystems of the past as well. Unfortunately, our knowledge about plankton composition of the past is still quite limited. As an important part of today’s zooplankton, malacostracan larvae are still a rarity in the fossil record; many types of malacostracan larvae dominating the modern plankton have so far not been found as fossils. Here we report a new type of fossil malacostracan larva, found in the 150 million years old lithographic limestones of southern Germany (Solnhofen Lithographic Limestones). The three rather incomplete specimens mainly preserve the telson. A pronounced middle spine on the posterior edge of these specimens indicates that they are either larval forms of a clawed lobster or of an axiidean lobster, or of a closer relative to one of the two groups. The tergo-pleura are drawn out into distinct spines in one specimen, further supporting the interpretation as a larva of a clawed lobster or an early relative. The telson morphology also shows adaptations to a prolonged planktic life style, the latero-posterior edges are drawn out into distinct spines. Similar adaptations are known in larvae of the modern homarid lobster Nephrops norvegicus, not necessarily indicating a closer relationship, but convergent life styles. The new finds provide an important new insight into the composition of Mesozoic zooplankton and demonstrate the preservation potential of lithographic limestones.

Introduction

Plankton describes the entirety of organisms floating in the water column without the ability to actively swim against the current. Plankton has been recognised as an important basis for marine ecosystems in modern seas. Our understanding of changes in plankton composition are therefore important in modern conservation biology and ecology, but are also of economic importance (e.g., Duffy & Stachowicz, 2006; Torres et al., 2013; Sardet, 2015).

For long-time comparisons of changes in plankton composition, data of fossil plankton is of major interest. Plankton must have been similarly important in the past as it is today. Unfortunately, our knowledge of plankton in the past is often limited to very specific groups of organisms, namely those with strongly sclerotised and/or mineralised morphological structures, or those composed of certain decay-resisting substances such as resting cysts and the like (examples in, e.g. Lipps, 1970; Tappan & Loeblich, 1973; Leckie, 2009), while such groups dominating the modern plankton (often lacking easily preservable structures) have a very scarce or absent fossil record (e.g., Signor & Vermeij, 1994; Rigby & Milsom, 2000; Perrier, Williams & Siveter, 2015). It is nevertheless already possible to recognise that the plankton composition was quite different in the past (e.g.,  Nützel & Frýda, 2003; De Baets et al., 2012; Ritterbush et al., 2014; Servais et al., 2015).

In modern seas, crustaceans are a major part of the zooplankton. Among these we especially need to mention copepod crustaceans and larval stages of malacostracan crustaceans. Copepods are mainly a part of the microplankton, while decapod larvae additionally include forms of meso- and macroplankton.

So what about the fossil record of these groups? Fossil copepods are extremely rare. They are limited to only few instances (Palmer, 1960; Cressey & Patterson, 1973; Cressey & Boxshall, 1989; Bennike, 1998; Huys et al., 2016), often only very incomplete (Selden et al., 2010) or indirect findings (Radwańska & Radwański, 2005; Radwańska & Poirot, 2010). Malacostracan larvae are also rare in the fossil record, but new forms have been repetitively identified in recent years (e.g., Haug, Haug & Ehrlich, 2008; Haug et al., 2011a; Haug, Ahyong & Haug, 2014; Haug, Martin & Haug, 2015; Haug, Wiethase & Haug, 2015; Haug et al., 2015; Hyžný, Haug & Haug, 2016) and are currently limited to the Mesozoic. Despite their rarity, each of these finds marks another important data point for our understanding of plankton in the past.

Due to preservation biases, large larval forms appear to be more commonly found as fossils, not necessarily representing the original composition of the fauna, or the true diversity. The most commonly found fossil malacostracan larvae are consequently the super-sized larvae of spiny lobsters and slipper lobsters (with up to 150 mm in the extant fauna). These are represented by at least a dozen different forms (Polz, 1984; Polz, 1995; Polz, 1996; Haug et al., 2013a; Haug & Haug, 2016), some known from thousands of individuals (Polz, 1971; Polz, 1972; Polz, 1973) and occurring in at least three Lagerstätten (Polz, 1984; Pasini & Garassino, 2009; Tanaka et al., 2009; Haug et al., 2011a). Other more uncommon fossils are also giant larval froms like those of the raptorial mantis shrimps (Haug, Haug & Ehrlich, 2008; Haug et al., 2009; Haug et al., 2010; Haug, Ahyong & Haug, 2014; Haug, Wiethase & Haug, 2015) or the polychelidan lobsters (today only represented by a relic group, mostly restricted to the deep sea; Haug et al., 2015; Eiler, Haug & Haug, 2016). Groups of larvae dominating the modern plankton, like larvae of true crabs, false crabs and their relatives (Meiura = Brachyura and Anomala, the latter also called Anomura), are very rare on the other hand (Luque, 2015; Haug, Martin & Haug, 2015; Hyžný, Haug & Haug, 2016), as their larvae are significantly smaller and more unlikely to be preserved.

Brachyuran and anomalan crustaceans are “still on their way” in the Mesozoic, only diversifying in the later Mesozoic (see discussion in Haug, Martin & Haug, 2015). It is therefore not to be expected that their larvae were as abundant as in modern oceans. Instead the lobster-like crustaceans, especially polychelidan and clawed lobsters (and their relatives) were dominating parts of the faunas as adults, especially in the mid-Mesozoic, hence the Jurassic. We should therefore expect that larval forms of these groups have represented important parts of the Jurassic plankton. Yet, so far no definite fossils of such planktic larvae have been identified. Possible late larvae of Jurassic polychelidans are late transitory stages, not the truly planktic ones (Eiler & Haug, 2016). Some fragmentary specimens have been discussed as possible remains of small malacostracans (Haug et al., 2011a; Haug, Ahyong & Haug, 2014) and might well be parts of larvae of clawed lobsters, but due to the limitations of preservation this must remain speculative.

Here we report the first definite finds of zoea-type larvae of clawed lobsters, hence truly planktic larval forms of this group. We discuss the impact of this new finding on reconstructing the plankton composition in the past and how this finding influences our strategies for detecting further material of fossil malacostracan larvae.

Material and Methods

Material

Three specimens were available for this study (Fig. 1). All originate from the private collection of Roger Frattigiani, Laichingen, and are now deposited in the Staatliches Museum für Naturkunde Stuttgart (SMNS 70353/1, 70353/2, 70353/3). Specimens were originally found in the Birkhof quarry in the Blumenberg area near Eichstätt (Solnhofen Lithographic Limestones, Upper Jurassic, Tithonian, Hybonotum zone, Riedense subzone; Schweigert, 2007). For comparison an extant albuneid zoea from the collections of the Muséum national d’Histoire naturelle Paris was documented (MNHN IU-2014-5527). High-resolution images of the specimens are available at https://www.morphdbase.de/ (C_Haug_20161213-M-5.1, C_Haug_20161213-M-6.1, C_Haug_20161213-M-7.1, C_Haug_20161213-M-9.1).

Figure 1 Entire material of larval specimens, composite-fluorescence micrographs.

All three specimens shown in the same scale to allow relative size comparison. Matrix digitally removed for clarity.

Documentation method

All specimens were documented on a Keyence BZ-9000 inverse epifluorescence microscope, exploiting the autofluorescence of the fossils (Haug et al., 2011b). Blue–green fluorescence (GFP) was used instead of the commonly used UV fluorescence (e.g., Tischlinger & Arratia, 2013). UV fluorescence is often unfortunate due to dust, which shows a very strong fluorescence and is in many cases not removable from the fossils. This is especially true for the comparably high magnifications as applied here (4 × objective lens, resulting in 40 × magnification). Due to the magnification, depth of field and field of view were limited. To overcome these limitations, several stacks of images for several adjacent image details were recorded (see details Haug, Haug & Ehrlich, 2008). Additionally, in some areas where the fluorescence capabilites differed strongly, two stacks were recorded, each with a different exposure time (Haug et al., 2013b).

Image processing

Stacks of images were fused to sharp images with CombineZP. Fused images were stitched to panoramas using the photomerge function of Adobe Photoshop CS3 or Elements 11. Images of different exposure times were combined into a single evenly illuminated image following the procedure described in Haug et al. (2013b).

Results

Specimen 1 (SMNS 70353/1)

Specimen 1 is most complete, but still largely represents fragmentary remains (Fig. 2). The overall colour and texture of the surface already clearly indicate that these fragments are the remains of a crustacean (although these characters are hard to quantify, they are significantly different to those in co-occurring groups such as insects, echinoderms or fishes, or also in strongly calcified crustaceans such as isopods). This is also in concordance with the preserved structures.

Figure 2 Specimen 1 (SMNS 70353/1), composite-fluorescence micrographs.

(A) Overview of entire specimen, although incomplete showing general organisation; arrows mark small spines. (B) Close-up on posterior rim of telson; arrowheads point to small hair-like structures or setae. Abbreviations: fl?, possible flagellum of antennula or antenna; hs, head shield; ms, median spine; pls, postero-lateral spine; ps3, 5, pleon segment 3, 5; pt, pleotelson; rs, rostrum; sp, spine; ur, uropod.

Most anteriorly a shield structure is apparent. It appears to be embedded in a dorso-lateral orientation. The anterior rim is drawn out into a distinct but stout rostrum. Along the edge at least two spines are apparent (Fig. 2A). Close to the shield an elongate structure is preserved composed of five elements. Further distal elements are narrower than proximal ones. The structure most likely represents the flagellum of an antennula or antenna. Close to the posterior part of the shield, a piece of rectangular outline is apparent, most likely representing an isolated element of one of the trunk appendages (posterior thoracopods, “pereiopods”).

The posterior trunk (pleon) is incompletely preserved and an isolated piece is interpreted as the tergite of pleon segment 3. It is domed and the latero-posterior edges are drawn out into distinct spines. Medially along the posterior rim a posteriorly pointing spine is apparent.

The next posterior preserved piece resembles the tergite of pleon segment 3 in overall morphology and is interpreted as the tergite of pleon segment 5. It is slightly larger than the tergite of pleon segment 3, also the spines are more pronounced.

Articulated to pleon segment 5 is an elongated part posterioly extending into a more or less triangular structure. This is interpreted as a compound part of pleon segment 6 and the telson, hence a pleotelson. The posterior part of the pleon segment is partly twisted, and folded onto itself. Still the triangular outline (in dorsal view) of the telson is apparent. The posterior rim bears a prominent median spine (Fig. 2B). Left and right to it numerous hair-like structures are apparent. It remains unclear whether these are jointed (true setae) or not (trichomes). The latero-posterior corners are drawn out into elongate distally tapering spines. Close to the base of each large spine, on its median side, slightly laterally from the hair-like structures is a smaller spine, about the same length as the hair-like structures but more massive.

Specimen 2 (SMNS 70353/2)

In comparison to specimen 1, specimen 2 is clearly identified as an isolated telson (Figs.3A–3D). It strongly resembles the posterior part of the pleotelson of specimen 1, but is not twisted and therefore provides additional structural information. The overall size is similar to that of specimen 1. The outline is strongly triangular in dorsal view. The anterior edges, most likely marking the transition to pleon segment 6 are marked by a pair of laterally extending small spines (Fig. 3A).

Figure 3 Specimens 2 (SMNS 70353/2) and 3 (SMNS 70353/3), each representing an isolated posterior part of a pleotelson (=telson) composite-fluorescence micrographs.

(A)–(D) Specimen 2. (A) Overview; arrows mark small spines. (B) Close-up on left lateral rim of telson; arrow points to small spine. (C) Close-up on right lateral rim of telson. (D) Close-up on left posterior rim of telson. (E–G) Specimen 3. (E) Overview; arrow marks small spines. (F) Close-up on right lateral rim of telson. (G) Close-up on left posterior rim of telson. Abbreviations: ms, median spine; pls, postero-lateral spine; set, hair-like structures or setae; sp, spine; ur, uropod.

Postero-lateral edges are drawn out into massive spines, forming a shallow angle. The number of hair-like structures along the posterior rim is 13 per side (Fig. 3D). They are all roughly the same length and the distances between them appear evenly distributed.

Remains of the uropods appear to be preserved left and right to the telson (Figs. 3B and 3C). Most likely these remains represent the outer, stronger sclerotised edges of the exopods. One small spine appears to be preserved close to the distal end on the posterior surface of the exopod.

Specimen 3 (SMNS 70353/3)

This specimen strongly resembles specimen 2 and is therefore also interpreted as an isolated telson (Figs. 3E–3G). Size, general morphology and number of structures are all similar to specimen 2. Yet, it is not as complete; for example, the posterior edge right to the median spine is broken. It differs from specimen 2 only in the angle of the posterior spines. These form a much narrower angle, as they point less far laterally, but more posteriorly.

Discussion

Systematic interpretation

The specimens are considered conspecific, but differ in the angle between the postero-lateral spines. As all specimens have a similar overall size of the telson region (Fig. 1), it seems unlikely that this difference is a ontogenetic one indicating the presence of several instars. By comparison to modern forms it seems most likely that the spines originally had a certain flexibility and that the difference in angle reflects a preservational difference. We therefore see no possibility to diagnose different forms and see conspecifity as the most parsimonious explanation.

The overall morphology of the specimens indicates that they represent larval malacostracans. An important character in this aspect is the pronounced middle spine of the posterior edge of the telson. In adult malacostracans the telson is often elongate triangular in dorsal view, but with the tip pointing posteriorly, or rectangular to square-shaped in dorsal view. In many larval forms, for example, in decapods, the telson appears forked with a pronounced median indent or, similar to the adults, rectangular, with an evenly armed posterior edge (Martin, Olesen & Høeg, 2014). Forward pointing triangular to tapezoid/trapezium telson shapes with a pronounced median spine occur in modern forms only in larvae of nephropid or axiidean lobsters (Fig. 4). The latter seem to lack tergo-pleura drawn out into posteriorly pointing spines on the pleon (Dos Santos & González-Gordillo, 2004; Pohle & Santana, 2014). As such spines are present in one of the fossils and in modern nephropid lobsters (Jorgensen, 1925; Wear, 1976; Smith, 1987; Goy, 2014), the fossil larvae most likely represent larvae of clawed lobsters, i.e., nephropids or now extinct relatives of them.

Figure 4 Comparison of the new fossil larva with extant forms.

New fossil larva as restoration in dorsal aspect; each of the others as an isolated telson in dorsal view. Homarus gammarus (European lobster); zoea III simplified from Rötzer & Haug (2015). Nephrops norvegicus (scampi, Kaisergranat); zoea III combined from Smith (1987) and Jorgensen (1925). Metanephrops challengeri; late zoea simplified from Wear (1976). Axius stirynchus; late zoea simplified from Dos Santos & González-Gordillo (2004, Fig. 2F). Undetermined albuneid larva; late zoea simplified from Fig. 5.

Figure 5 Extant larva of an albuneid meiuran for functional comparison; cross-polarised macro images.

Note the long postero-lateral spines on shield and telson and the triangular telson. (A) Lateral view on left side. (B) Dorsal view on head shield. (C) Posterior view on anterior region; dorsal view on posterior pleon and pleotelson; arrows mark spines. (D) Detail of the telson. Abbreviations: hs, head shield; pls, postero-lateral spine; pt, pleotelson; set, hair-like structures or setae.

The difficulties with ‘clawed lobsters’

Modern clawed lobsters comprise the true lobsters and the reef lobsters, yet quite a number of fossil forms also resembles clawed lobsters, such as erymid or glypheid lobsters, in general habitus (e.g., Garassino & Schweigert, 2006; Charbonnier et al., 2013; Charbonnier et al., 2015; Bracken-Grissom et al., 2014; Hyžný et al., 2015; Schweitzer et al., 2016). The exact relationship of these groups remains still partly unclear (see recent discussions in Charbonnier et al., 2015; Hyžný et al., 2015). As also axiidean lobster larvae possess a pronounced median spine on the posterior edge of the telson it is possible that this feature characterises a larger group including erymid and glypheid lobsters. Therefore, we can currently not be more precise with the systematic interpretation of the larvae described herein. They may represent larval forms of nephropid, erymid or glypheid lobsters or a form closely related to them.

Functional comparison

The telson of the new fossil larvae clearly shows adaptations to a prolonged life in the pelagic realm. Such a life style requires morphological specialisations for staying in the water column without loosing much energy, which means that the buoyancy needs to be enhanced (e.g., Perrier, Williams & Siveter, 2015). This is especially important if the larvae reach relatively large body sizes such as the specimens investigated here (although they are not as large as those of polychelidan or achelatan lobsters).

The investigated specimens bear pronounced latero-posterior spines on the telson, with this distantly resembling the telson of larvae of Nephrops norvegicus (scampi). Here also the latero-posterior corners of the telson are strongly drawn out into spines, even more than in the fossils (Fig. 4). As other nephropids, N. norvegicus larvae also possess a median spine on the posterior edge. The telson of N. norvegicus larvae is also triangular in dorsal view, yet not as pronounced as in other nephropids (Jorgensen, 1925; Smith, 1987).

The overall morphology of the telson of the fossils additionally shows an overall similarity to the larvae of certain false sand crabs (Albuneidae; e.g., Knight, 1970; Stuck & Truesdale, 1986; Harvey et al., 2014), besides the fact that these lack the median spine (Figs. 4 and 5). Still the overall shape is triangular, the corners are drawn out into spines and the posterior edge is armed. We can assume that the specialised telson of the fossil larvae provided additional buoyancy for the rather large larvae, similar to albuneid larvae.

Significance

The fossil record of arthropod zooplankton appears to be very incomplete (Perrier, Williams & Siveter, 2015). In general, the overall reconstruction of fossil zooplankton seems based largely on estimations, as especially the microplankton is in focus of different studies, while the mesoplankton is usually less studied. Larvae are in such approaches rarely treated in detail (Rigby & Milsom, 2000), and, although the fossil record of malacostracan larvae is growing, it appears to be generally regarded as virtually absent (see recent review of Perrier, Williams & Siveter, 2015)

Yet, arthropods do play an important role in the modern plankton, especially the larval stages of malacostracans. As pointed out above, clawed lobsters and their relatives, i.e., nephropid, erymid and glypheid lobsters represent an important part of the marine benthic fauna in the Mesozoic. We should therefore expect that their larvae are a major share of the plankton of that time. Finding such larvae is thus important for corroborating this assumption.

As discussed above, the newly described specimens show adaptations for prolonged life in the plankton and therefore will represent the upper threshold of size for such larvae. Other larvae of clawed lobsters will be significantly smaller. The larvae furthermore most likely represent only a single specialised form of a wider range of different types of larvae. It has been demonstrated that zoea-type larvae of achelatan lobsters were morphologically more diverse than the larvae of modern forms (Haug et al., 2013a). We can expect that clawed lobster larvae also were morphologically more diverse, possibly similarly diverse to larvae of modern meiuran forms.

The fossils demonstrate that it is possible to find such important components of the plankton and also give an important hint what to look for. Haug et al. (2011a) suggested that some incomplete remains represent isolated shields of larvae, as these might have had a higher preservation potential. The fossils described herein show that quite the other end of such a larva, the telson, might also have high preservation potential. Focused search for such remains should provide additional insights into the plankton composition of the past.

We would like to thank Roger Frattigani, Laichingen for providing the specimens. Gideon T. Haug, Neuried photographed the specimens, for which we are very grateful. Laure Corbari, MNHN Paris, kindly provided access to the extant crustacean collections. Kenneth De Baets, Erlangen, is thanked for helpful comments and for editorially handling the manuscript. Vincent Perrier, Lyon, Carrie Schweitzer, Kent, and Matúš Hyžný, Bratislava, are heartily thanked for their comments. J. Matthias Starck, München is thanked for support and discussions. We thank all people providing free and low-cost software.

Additional Information and Declarations

Competing Interests

Author Contributions

Data Availability

The authors declare there are no competing interests.

Joachim T. Haug conceived and designed the experiments, performed the experiments, analyzed the data, wrote the paper, prepared figures and/or tables, reviewed drafts of the paper.

Carolin Haug performed the experiments, analyzed the data, wrote the paper, reviewed drafts of the paper.

The following information was supplied regarding data availability:

MorphDBase:

C_Haug_20161213-M-5.1; C_Haug_20161213-M-6.1; C_Haug_20161213-M-7.1; C_Haug_20161213-M-9.1.

www.morphdbase.de/?C_Haug_20161213-M-5.1;

www.morphdbase.de/?C_Haug_20161213-M-6.1;

www.morphdbase.de/?C_Haug_20161213-M-7.1;

www.morphdbase.de/?C_Haug_20161213-M-9.1.

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
