# Peer review of "A new glimpse on Mesozoic zooplankton—150 million-year-old lobster larvae"

_PeerJ, doi:10.7717/peerj.2966_

## Round 0.1 · original submission · Minor Revisions

· Academic Editor

Minor Revisions

Thank you for submitting this interesting work on lobster larvae to PeerJ which is attested by unanimous positive and enthusiastic assessments by the reviewers and myself. However, some minor points still need to be addressed before publication.

The main points to be addressed are:

Preservation: As pointed out by reviewer 2, you state that the colors and texture already indicate that you are dealing with crustaceans. I concur that crustaceans do have a particular texture and color in “Plattenkalken”, but maybe you could explain this in a little more detail for people not familiar with this phenomenon.

Taxonomy: According to Reviewers 2 and 3, part of the common names and taxonomy used are considered outdated. Much more recent papers have discussed the issues among and relationships between clawed lobster groups (See references listed by Reviewers 2 and 3). Comparing your material with Axiidea and Gebiidea might also be worthwile based on the morphology of the telson according to both reviewers. Reviewer 2 also point out that comparison of your specimens with a larvae of “clawed larvae” might be more suitable than comparing them with anomuran larvae (if possible).

Functional Comparison: Reviewer 2 does not really agree that the telson is similar to Nephrops norvegicus, which might extend the range of the enclosing family Nephropidae beyond the Cretaceous. Reviewer 2 feels that morphological comparisons with Axiidae or Gebiidae or other “clawed lobster” families (see also comments by Reviewer 3), which are also known from older rocks, might be more appropriate than comparing them with anomuran larvae.

Formatting: Some references are missing from reference list and/or text or inconsistently formatted (see comments by Reviewer 3).
In addition to the reviewer comments, my suggestions listed in the annotated pdf should also be addressed. Some abbreviations or explanations are missing from figure captions (see comments by Reviewer 2).

·

Basic reporting

no comments

Experimental design

no comments

Validity of the findings

No comments

Comments for the author

This paper represent an important contribution to the general knowledge of fossil planktonic arthropods and I strongly recommend its publication after some minor editing (see attached pdf).

·

Basic reporting

The paper is generally fine. There are some issues of clarity that should be addressed. Anomura/Anomola are not really called false crabs, as they include hermit crabs, squat lobsters, and other animals. An Anomuran larvae is illustrated for comparison to fossil larvae that are considered to belong to "clawed" lobsters. I suggest illustration of larvae of clawed lobsters to make this comparison much more robust.
Under the description and reporting for Specimen 1, it is stated that "The overall color and texture of the surface already clearly indicate that these fragments are the remains of a crustacean." I am not sure what colors or textures indicate that animals are crustaceans in the fossil record, nor do I think that non-crustacean specialists would know this either. Please explicitly state what it is about these specimens that suggest that they are crustaceans.
Under difficulties with "clawed lobsters," Garassino and Schweigert" are cited as a relevant reference. Much more recent papers have discussed the issues among and relationships between clawed lobster groups including those by Charbonnier et al., Feldmann et al., and Schweitzer et al.
Thalassinideans as used in recent literature refers only to one family Mud and ghost shrimp are now placed within Axiidea and Gebiidea. It seems based upon the morphology that the telson could belong to members of these two latter groups as well.
Under Functional Comparison, I don't really think that the telson is that similar to Nephrops norvegicus. The enclosing family, Nephropidae, is known from the early Cretaceous to Recent so this occurrence would extend the range of the family, which should be noted. Axiids and Gebiids are known from older rocks as are other "clawed lobster" families.
Figure 2 is missing an explanation for the abbreviation rs. In figure 5, the term meiuran is used without explanation. Non-specialists will not know what it means.

Experimental design

Techniques are well-explained, executed, and creative.

Validity of the findings

Findings are certainly valid, and finding more larvae in the fossil record is very important. The paper needs to be cleaned up and updated to some more recent literature.

Comments for the author

I think it looks more like the axiid/gebiid that is illustrated or the Metanephrops. Perhaps functional comparison with those two groups would be more appropriate.

·

Basic reporting

List of references need uniform format.
Some literature items in the list of references are not mentioned in the text.
English can be improved on several places.

Experimental design

No comments.

Validity of the findings

I recommend not to use "thalassinidean lobsters", instead "axiideans" should be used. Discussion on some minor aspects can be expanded or at least, some references can be added (all details are mentioned in the annotated manuscript PDF).

Comments for the author

This is a valuable contribution on fossil decapod larvae. I fully recommend its publishing - after minor revision. All my comments are mentioned in the annotated manuscript PDF.

---

## Round 0.2 · Minor Revisions

· Academic Editor

Minor Revisions

Thank you for addressing all points raised by the reviewers and myself. The manuscript is as good as accepted. I just found some minor additional things in the revised version i would like you to address before publication.

Line 19: i guess you mean "a close relative of one of these groups" rather than a "closer relative to one of the two groups", please check.

Line 39: i think you need to add something additionally here as sclerotization mostly refers to hardened animal structures, but not to sporopollenin and other resistant compounds building up unicellular or "phytoplankton". In this case, it might help to refer to resting cysts and the like.

Line 263-264: i might be good to separate the reviewers from the editor in your listing in the acknowledgements for completeness sake...

---

## Round 0.3 · accepted · Accept

· Academic Editor

Accept

Thank you for implementing these final suggestions and remarks.